# Pulmonary MicroRNA Changes Alter Angiogenesis in Chronic Obstructive Pulmonary Disease and Lung Cancer

**DOI:** 10.3390/biomedicines9070830

**Published:** 2021-07-16

**Authors:** Clara E. Green, Joseph Clarke, Roy Bicknell, Alice M. Turner

**Affiliations:** 1Institute of Inflammation and Ageing, College of Medical and Dental Sciences, University of Birmingham, Edgbaston, Birmingham B15 2TT, UK; 2Institute of Cardiovascular Sciences, College of Medical and Dental Sciences, University of Birmingham, Edgbaston, Birmingham B15 2TT, UK; JJC946@student.bham.ac.uk (J.C.); r.bicknell@bham.ac.uk (R.B.); 3Institute of Applied Health Research, College of Medical and Dental Sciences, University of Birmingham, Edgbaston, Birmingham B15 2TT, UK; a.m.turner@bham.ac.uk

**Keywords:** pulmonary endothelium, COPD, lung cancer, miRNA, microarray, angiogenesis

## Abstract

The pulmonary endothelium is dysfunctional in chronic obstructive pulmonary disease (COPD), a known risk factor for lung cancer. The pulmonary endothelium is altered in emphysema, which is disproportionately affected by cancers. Gene and microRNA expression differs between COPD and non-COPD lung. We hypothesised that the alteration in microRNA expression in the pulmonary endothelium contributes to its dysfunction. A total of 28 patients undergoing pulmonary resection were recruited and endothelial cells were isolated from healthy lung and tumour. MicroRNA expression was compared between COPD and non-COPD patients. Positive findings were confirmed by quantitative polymerase chain reaction (qPCR). Assays assessing angiogenesis and cellular migration were conducted in Human Umbilical Vein Endothelial Cells (*n* = 3–4) transfected with microRNA mimics and compared to cells transfected with negative control RNA. Expression of miR-181b-3p, miR-429 and miR-23c (all *p* < 0.05) was increased in COPD. Over-expression of miR-181b-3p was associated with reduced endothelial sprouting (*p* < 0.05). miR-429 was overexpressed in lung cancer as well and exhibited a reduction in tubular formation. MicroRNA-driven changes in the pulmonary endothelium thus represent a novel mechanism driving emphysema. These processes warrant further study to determine if they may be therapeutic targets in COPD and lung cancer.

## 1. Introduction

Chronic obstructive pulmonary disease (COPD) and lung cancer are highly prevalent smoking-related conditions. COPD is a condition characterised by airflow obstruction, which is not normally fully reversible with the use of bronchodilators and is generally thought to be progressive over time [1]. Damage in both the airways and lung parenchyma contribute to the airway obstruction [1,2]. Spirometry is used to diagnose airway obstruction, which is defined as a ratio of Forced Expiratory Volume in one second (FEV1) to Forced Vital Capacity (FVC; the total volume expired) of less than 0.7 [3]. COPD is common and has been predicted to be the third leading cause of death by the World Health Organization (WHO) by 2030 [4]. There are over 2 million cases of lung cancer each year and survival rates have changed relatively little over time despite advances in treatment [5]. Focusing lung cancer screening on high-risk groups could identify patients early and thus raise cure rates. Patients with COPD are one such group; the incidence of lung cancer in COPD patients is higher than in healthy ones, even after adjustment for smoke exposure [6], suggesting a possible shared pathogenesis. By understanding the pathogenesis of COPD and lung cancer in detail, it is possible that new treatments may be developed and the risk of lung cancer in COPD may be reduced.

The endothelium has long been known to play a role in COPD. In the 1950s, Liebow demonstrated that the alveolar septa in COPD patients were almost avascular, leading to the hypothesis that vascular atrophy resulted in alveolar destruction [7]. Supporting this concept, increased levels of apoptotic endothelial cells have been identified in the lungs of patients with COPD [8]. Pulmonary endothelial cells also show signs of injury in COPD, with increased expression of oxidative stress–advanced glycation end products (AGEs) [9]. Previous researchers have shown that it is possible to induce emphysema in rodents by deliberately causing endothelial apoptosis [10]. A reduction in the endothelium in patients with emphysema may be caused by reduced levels of Hypoxia Inducible Factor-1α (HIF-1α) and Vascular Endothelial Growth Factor (VEGF). Levels of HIF-1α and VEGF appear to be related to COPD disease severity: both are correlated with FEV1 percentage predicted in patients with emphysema [11].

In addition to altered levels of endothelium in patients with COPD, the endothelium appears to behave in a dysfunctional manner. Endothelial dysfunction is defined as disturbed endothelial-dependent vasodilatation [12]. It results in a breakdown of the microvascular endothelial barrier and loss of the anti-adhesive and anti-thrombotic functions of the endothelium [12]. Endothelial dysfunction is associated with the severity of COPD and is related to FEV1 and the percentage of emphysema on CT scan [13,14]. These associations are independent of smoking and other major causes of endothelial dysfunction [14,15]. The relationship between endothelial dysfunction and FEV1 is explained by the percentage of emphysema [14]. This suggests that endothelial dysfunction might be involved in emphysema pathogenesis and COPD [14]. Endothelial dysfunction is also related to clinical outcomes: patients with increased endothelial dysfunction have reduced 6 min walk test results and a worse overall prognosis with increased exacerbation rates [16,17,18].

MicroRNAs (miR) are small (around 20 nucleotides) non-coding ribonucleic acids (RNAs) that regulate gene expression [19]. Human studies suggest a role for miR in COPD pathogenesis since miR expression differs between patients with and without COPD [20]. However, very few studies have looked at miR expression in individual cell types; this is important because miR changes may not apply to all cells, such that analysis of the whole lung may mask signals present in one cell type and obscured by opposite signals in another. Furthermore, miR changes in COPD patients may precede the development of lung cancer. Bronchoalveolar lavage (BAL) samples from people with COPD have been shown to have deregulated miR common to samples from patients with adenocarcinoma [21]. There is limited evidence about the role of endothelial miR in COPD. However, cigarette smoke, the main pathogenic factor for COPD, alters the expression of miR in endothelial microparticles (EMPs) in humans [22]. Changes in miR expression seen in COPD can also suppress HIF-1α expression in pulmonary endothelial cells and so may contribute to COPD pathogenesis as above [23]. Supporting this, another study identified that miR-34a was upregulated in Human Pulmonary Endothelial Cells (HPECs) exposed to smoke and that this resulted in apoptosis through Notch-1 suppression [24]. We therefore chose to study the miR profile of the pulmonary endothelium in COPD. Targets identified as upregulated in COPD were then sought in lung cancer in order to identify shared mechanisms of pathogenesis.

## 2. Materials and Methods

### 2.1. Patient Recruitment

A total of 28 patients undergoing thoracic surgery between August 2013 and November 2014 were included in the study. All were recruited as part of the Midlands Lung Tissue Consortium (MLTC) and gave informed consent. The MLTC was approved by The Health Research Authority National Research Ethics Service (07/MRE08/42). Most patients were undergoing lobectomy or pneumonectomy for suspected lung cancer, although those undergoing lung volume reduction surgery were also approached. Baseline data, including age, sex, body mass index (BMI), pack years, lung function (FEV1, FVC and other measures if available), presence of chronic bronchitis and emphysema, and tumour stage, were obtained from clinical records, including CT scan images and pathological reports. 

### 2.2. Tissue Collection and Extraction of Human Pulmonary Endothelial Cells (HPECs)

Healthy lung tissue was obtained from lobectomy or pneumonectomy samples as far away from the tumour site as possible. The lung tissue was divided into samples approximately 3 cm long, chopped into fine pieces and digested using collagenase V. Following digestion, the lung solution was filtered and the endothelium isolated by positive selection using *Ulex Europaeus* lectin-coated magnetic beads, as described in our previous work [25], prior to storage in Trizol at −80 °C. This process was repeated with tumour tissue where available in both COPD and non-COPD patients to obtain tumour endothelium (*n* = 8). An overview of the method is illustrated in Figure 1A. qPCR to look for CD31 enrichment in endothelial isolates was used after extraction to confirm a pure population of endothelial cells, as described previously (Figure 1B) [25].

### 2.3. Processing of RNA and Conduct of Arrays

RNA was extracted from endothelial samples using the Qiagen (Germantown, MD, USA) miRNeasy Mini Kit according to the manufacturer’s instructions. RNA electrophoresis was used to confirm RNA integrity sufficient for arrays, as defined by an RIN > 5 (Agilent 2100 Expert Software (Santa Clara, CA, USA)). Genomewide miR arrays (Agilent Sureprint Human miRNA microarrays) were processed by a core facility at the University of Birmingham. One patient cohort was analysed for miR expression (*n* = 13, 7 non-COPD vs. 6 COPD) using Significance Analysis of Microarrays (SAM) [26]. Results were limited to a fold change of 2 as this represents a large, biologically significant difference in expression between groups [27]. Genes likely to be targeted by miR and significantly differently expressed between groups were ascertained using DIANA-microT-CDS and TargetScan [28,29,30]. All array data have been deposited in the ArrayExpress database at EMBL-EBI (www.ebi.ac.uk/arrayexpress, accessed on 3 March 2021) under accession number E-MTAB-10311 [31].

### 2.4. Validation of Array Data

Validation of positive microarray results was performed using qPCR as per our previous work [25]. MiR which differed in the arrays were confirmed using qPCR; initially, the TaqMan MicroRNA Reverse Transcription Kit (Applied Biosystems, Waltham, MA, USA) was used to convert to cDNA. TaqMan microRNA assay kits were used as per manufacturer’s instructions. All qPCR reactions were run in triplicate, with expression of each miR normalised to RNU48 to calculate the delta CT (∆CT).

### 2.5. miR Overexpression

MiR mimics (Qiagen miScript, Qiagen) were transfected into Human Umbilical Vein Endothelial Cells (HUVECs; Lonza, Basel, Switzerland). HUVECs were chosen as they are well-established in endothelial models and have been previously used in the investigation of miR in lung disease [25,32,33]. HUVECs were plated on to 0.1% gelatin-coated plates the day before transfection and grown in HUVEC media (see Appendix B) to obtain near-confluent HUVEC cultures. On day two, transfection was performed. Transfection was performed using mimics at a final concentration of 10 nM and lipofectamine (Lipofectamine^®^ RNAiMAX Transfection Reagent, ThermoFisher Scientific, Waltham, MA, USA; 13778075) at a concentration of 0.3% by diluting with optimem (ThermoFisher Scientific). The cells were left in the incubator at 37 °C, 5% CO_2_ for 4 h to allow time for transfection to occur. Successful transfection was assessed using qPCR as above. All mimic experiments were conducted alongside negative siRNA (AllStars Negative Control siRNA, Qiagen), Optimem and Lipofectamine controls and were run in triplicate.

### 2.6. Cell Growth Assay

This was performed according to previous work [34]. First, 4 h after transfection, 2× trypsin (Gibco, Waltham, MA, USA) was used to remove cells, which were transferred into 15 mL falcon tubes. After washing, the cells were diluted to a concentration of 12,500 cells per mL. Then, 1 mL of cell solution (from each condition) was added to 3 wells of 3 0.1% gelatin-coated 12-well plates and placed into an incubator at 37 °C, 5% CO_2_. The following day, cells from 1 plate were removed, resuspended in 100 µL media and counted with a haemocytometer. This was repeated the next day for another plate and on the third day for the following plate. The numbers of cells from days 1–3 were compared for each condition to gain an estimate of cell growth during this time. All experiments were run in triplicate.

### 2.7. Cell Cycle Analysis

This was performed 48 h after transfection. Analysis of cell cycle was achieved by fixing cells in alcohol and staining with propidium iodide. Firstly, cells were washed before being added to 1 mL 85% ice-cold ethanol (VWR, Radnor, PA, USA) and stored at 4 °C. After fixation, cells were centrifuged at 200× *g* for 5 min and alcohol was removed. Cells were stained using 10 µL Propidium Iodide (Invitrogen, Waltham, MA, USA), 10 µL RNAse A (Qiagen) and 10 µL 10% Triton-X-100 (Sigma-Aldrich, St. Louis, MI, USA) in phosphate-buffered saline (PBS, Gibco). The cells were incubated at 37 °C, 5% CO_2_ for 20 min to ensure staining of cells. Cells were run through the CyAn flow cytometer (Beckman-Coulter, Brea, CA, USA) and the results were analysed using FloJo. Two cytogram plots were used to isolate the desired population of cells and to identify single cells only (Appendix A). A histogram of cell counts was used to identify the proportion of cells within each phase of the cell cycle. All experiments were run in triplicate.

### 2.8. Scratch Wound Assay

First, 4 h after transfection, 8000 endothelial cells were placed into each well of a 96-well ImageLock plate (Essen Biosciences, Ann Arbor, MI, USA) in HUVEC media. The cells were placed into the incubator at 37 °C, 5% CO_2_ for 48 h. A ‘Woundmaker 96’ (Essen Biosciences) was used to create the wounds in this experiment. The ImageLock plate was inserted into an Incucyte incubator at 37 °C, 5% CO_2_, which was programmed to take images at regular intervals using IncuCyteZoom 2015A software (Essen Biosciences). The images were downloaded and analysed using ImageJ. All experiments were run in triplicate.

### 2.9. Matrigel Assay

First, 48 h after transfection, 70 µL of Matrigel (Corning, Corning, NY, USA) was placed into a well of a 12-well plate and allowed to solidify by placing the plate into the incubator at 37 °C, 5% CO_2_ for 30 min. Cells were diluted to a concentration of 140,000 cells per ml in HUVEC media and 1 mL of cell solution was added to the Matrigel well. The plate was uploaded into an IncuCyte incubator (Essen BioSciences) at 37 °C, 5% CO_2_, which was programmed to take pictures at regular intervals using IncuCyteZoom2015A software (Essen BioSciences). The images were downloaded and analysed using the online available software ImageJ. All experiments were run in triplicate.

### 2.10. Spheroid Assay

Spheroid assays were conducted to assess endothelial sprouting, as previously described [35,36]. This was performed 48 h after transfection. Initially, endothelial cells were labelled with CFSE (Carboxyfluorescein succinimidyl ester). Cells in HUVEC media with methocellulose solution (see Appendix B) were added to each well of a 60-well microplate (Nunc, Roskilde, Denmark; 439225). The lid of the plate was attached, and the plate was inverted. The plate was incubated overnight at 37 °C, 5% CO_2_. Spheroids were harvested from the plate and were centrifuged.

Next, 10 µL sodium hydroxide (Sigma-Aldrich; S2770) was added to a collagen mix (see Appendix B), which was used to resuspend the spheroids, which were then placed on to a well of a 24-well plate. The plate was incubated at 37 °C, 5% CO_2_ for 10 min before adding 100 µL of EBM2 media (endothelial cell growth media, Lonza; CC-4542 + CC-5036) and incubating at 37 °C, 5% CO_2_ for 8 h. Then, 4% paraformaldehyde was added to fix the cells. Spheroids were imaged using a Zeiss 780 Zen confocal microscope (Oberkochen, Germany) and were analysed using the ImageJ ‘Spheroid Analysis’ plugin developed at the University of Birmingham [37]. Five spheroids were viewed per condition.

### 2.11. Static Co-Cultures

HUVECs (2 × 10^5^/well) were seeded into 12-well Falcon plates for 24 h prior to transfection. Cells were incubated for 48 h before treatment with or without 100 U/mL tumour necrosis factor-α (TNFα; R&D Systems, Abingdon, UK) for a further 4 h [38,39] immediately prior to the static adhesion assay. Venous blood was collected from healthy donors into EDTA tubes (Sarstedt, Leicester, UK). Neutrophils (PMN) were isolated by centrifugation on two-step histopaque density gradients as previously described [38,39]. 

Static adhesion assays were performed using phase-contrast digital microscopy [38,39]. Purified neutrophils were allowed to adhere and bind to HUVECs for 6 min, after which non-adherent cells were removed by washing twice with cell-free PBSA. Digitised recordings of 5–10 random fields were made 2 and 9 min after washing to assess neutrophil adhesion and transmigration, respectively. Images were analysed offline using Image-Pro Plus software (Media Cybernetics, Marlow, UK). At both 2 and 9 min, the number of cells that had transmigrated was evaluated and expressed as a percentage of those cells that had adhered. All experiments were run in triplicate.

### 2.12. Endothelial–Fibroblast Co-Cultures

Co-cultures were performed to assess endothelial tube formation as described previously [34]. Fibroblasts were plated at 3 × 10^4^ per well in a 12-well plate in 1 mL of DMEM-F12 with 10% FBS and incubated overnight at 37 °C, 5% CO_2_. On day 4, HUVECs were transfected. On day 5, HUVECs were plated at 3 × 10^4^ per well in 1 mL of HUVEC media on top of the fibroblast cultures. On day 11, cells were fixed with 1 mL 70% (*v*/*v*) ethanol chilled to −20 °C and incubated for 30 min at room temperature. Cells were washed with PBS before incubation with Anti-human CD31 (Dako Omnis) (Agilent) Clone JC70A in PBS, at 37 °C for 40–60 min. Cells were washed with PBS before incubation with Anti-mouse IgG (whole molecule)–Alkaline Phosphatase (goat polyclonal, Sigma-Aldrich A4656) at 37 °C for 40–60 min. Cells were washed before 500 μL of SigmaFAST BCIP/NBT substrate (Sigma-Aldrich; S8820) was added to each well and cells were incubated for 25 min at room temperature. Cells were imaged at 1.6 magnification using XLi-Cap software. Tubule formation was quantified using the AngioSys software as previously described [34]. Experiments were run in triplicate.

### 2.13. Statistical Analysis

SPSS v22 (SPSS Inc, Chicago, IL, USA) was used to perform standard parametric and non-parametric tests to compare COPD and non-COPD patients. Cell-based experiments were analysed using ANOVA, with Tukey’s test performed post-hoc to compare miR mimic to negative siRNA groups. Significance was assumed at *p* < 0.05. 

## 3. Results

### 3.1. Patient Characteristics Were Not Significantly Different between Groups Other than FEV1

A total of 13 patients were included in the miR microarray: seven patients with COPD and six patients without COPD (Table 1). There was no difference between groups in terms of sex, age, body mass index (BMI), pack years of smoking, current smoking status or co-existing lung malignancy. FEV1 percentage predicted was significantly reduced in the COPD group (64% vs. 96.8%, *p* = 0.008) as expected.

### 3.2. CD31 Expression Was Significantly Increased in Extracted Pulmonary Endothelial Cells

Prior to performing qPCR for miRNA or mRNA targets, it was necessary to confirm that the pulmonary endothelium was enriched in the samples extracted using Ulex-coated magnetic beads [25]. To do this, qPCR was performed to compare *CD31* expression in the endothelial isolates (*n* = 14) to the bulk remainder tissue (*n* = 6). Figure 1b illustrates the results of this qPCR experiment. The expression of *CD31* was significantly increased in the endothelial isolate over 2.5-fold (*p* = 0.012). It is worth noting that endothelial isolation using the same method has previously shown a greater enrichment for *CD31* than seen in this experiment (e.g., 15-fold in lung tumour). The reduced fold increase seen here is likely due to the fact that the lung is highly vascular and the proportion of endothelial cells in the lung is high (30%) [25].

### 3.3. MiR Expression in HPECs Is Significantly Different in Patients with COPD

Microarrays demonstrated the significantly different expression of miR in HPECs between patients with and without COPD. In total, 888 miR were significantly upregulated in COPD HPECs (full list in the Appendix A); the corresponding heatmap is shown in Figure 2.

The list of significantly upregulated miR was compared to the existing literature on miR expressed in endothelial cells. Six miR were chosen for validation by qPCR as these had been expressed previously in the endothelium [40,41,42,43,44,45] and were possible modifiers of endothelial mRNAs of interest, as predicted by TargetScan and DIANA-microT-CDS: miR-18b-3p, miR-181b-3p, miR-193b-5p, miR-23c, miR-342-5p and miR-429. qPCR comparing miR expression in COPD (*n* = 4) and non-COPD (*n* = 4) confirmed that 3 of the 6 miR targets were significantly upregulated in HPECs from patients with COPD: miR-181b-3p, miR-23c and miR-429 (Figure 2). 

Two targets were selected for further functional validation: miR-181b-3p (the target most significantly differentially expressed between groups) and miR-429 (the only target increased in both COPD and lung cancer—see Section 3.4).

### 3.4. MiR-429 Expression Is Increased in HPECs in COPD and Lung Tumour

In order to identify potential miR targets in both COPD and lung cancer miR, qPCR experiments were performed using RNA isolated from lung tumour HPECs. The expression of each miR in lung cancer (*n* = 6) was compared to the expression in non-COPD (*n* = 4). MiR-181b-3p and -23c were not expressed in the lung tumour samples. However, miR-429 was significantly increased in the lung tumour samples by 9-fold (Figure 2), thus suggesting that miR-429 could represent a shared target for COPD and lung cancer.

### 3.5. MiR Mimics Successfully Increased miR Expression Levels without Altering Cell Growth or Cell Cycle

To confirm that transfecting 10 nM miR mimics provided a stably increased level of miR mimic expression in HUVECs, three experiments were performed, confirming that miR-181b-3p expression was maintained up to three days after transfection (Figure 3A). Each experiment was repeated in quadruplicate. These experiments demonstrated that the expression of the miR mimic was still significantly higher in the transfected cells three days after transfection. The fold-change in miR after transfection is consistent with previous studies [32,46,47].

To determine whether or not miR mimic transfection affects endothelial cell growth, three separate experiments were performed. Briefly, HUVECs that had been transfected with miR mimic, negative siRNA, lipofectamine only or optimem only were plated into three wells (per condition) of three different 12-well plates. The number of cells per condition was counted 1–3 days post-transfection (Appendix A). There did not appear to be any difference in cell number in the miR mimic, negative siRNA or lipofectamine conditions in any of the three experiments, suggesting that miR overexpression does not have a significant effect on cell growth. However, the cell number was significantly increased in the optimem only group, which suggests that exposure to lipofectamine may impair cell growth.

To determine whether miR mimic transfection led to any changes in the cell cycle, flow cytometry after propidium iodide staining was performed. HUVECs that had been transfected with miR mimic, negative siRNA, lipofectamine or optimem only were compared. This was performed in three separate experiments. Appendix A demonstrates the proportion of cells in phases G1, S and G2 across all experiments. The proportion of cells in each phase did not differ between groups significantly, suggesting that overexpression of miR did not alter the cell cycle. 

### 3.6. MiR-181b-3p and MiR-429 Significantly Reduce Endothelial Tube Formation and Angiogenesis In Vitro

To determine the effect of miR-181b-3p and miR-429 on HUVEC tube formation, the level of tube formation in Matrigel was assessed in HUVECs that had been transfected with miR-181b-3p/miR-429 mimic, negative siRNA, lipofectamine only or optimem (Figure 3B,C). In both cases, a Tukey’s test was performed post-hoc to compare individual groups, which demonstrated that the numbers of nodes were significantly reduced in HUVECs transfected with miR-181b-3p/miR-429 mimic compared to negative siRNA (*p* < 0.001). 

To determine how miR-181b-3p overexpression in HUVECs alters endothelial sprouting, three spheroid assays were performed with the same conditions as the Matrigel experiments (Figure 3D). Tukey’s tests were performed in post-hoc analyses, which demonstrated that spheroids in the miR-181b-3p mimic group had reduced sprouting in comparison to the negative siRNA group, suggesting that miR-181b-3p overexpression significantly reduced endothelial sprout formation (*p* = 0.021). 

Co-cultures of HUVEC on a fibroblast matrix were performed in order to assess miR-181b-3p/miR-429 effects on angiogenesis using the same conditions as above (Figure 4). The main role of the fibroblasts is to secrete extracellular matrix and growth factors, resulting in endothelial cells forming anastomosing tubules over the course of 1–2 weeks. Co-culture assays look at true endothelial migration and sprouting angiogenesis; hence, this assay is more reflective of the in vivo situation than Matrigel assays [48]. Tukey’s test was used to compare groups, which demonstrated that the number of junctions and tubules was significantly reduced in both mimic groups in comparison to the negative siRNA group, suggesting that miR-429 and miR-181-3p reduce angiogenesis.

### 3.7. MiR-181b-3p and miR-429 Do Not Affect Wound Adhesion

To assess the effect of miR-181b-3p/miR-429 overexpression on endothelial migration and wound healing, the rate of closure of wounds in endothelial monolayers was assessed in HUVECs that had been transfected with miR mimic, negative siRNA, lipofectamine only or optimem (Appendix A). There was no significant difference between groups. Wound closure was significantly different between groups at the 12 h time point in the experiments with miR-181b-3p (Appendix A). A Tukey’s test was performed post-hoc to compare results between groups, which demonstrated that the optimem only group had a smaller percentage wound area remaining compared to the lipofectamine group (*p* = 0.015). This was the only significant difference detected. As wound closure did not vary significantly between the miR-181b-3p/miR-429 mimic and negative siRNA groups, these results suggest that miR-181b-3p and miR-429 do not have a significant effect on endothelial migration and wound closure. 

### 3.8. MiR-181b-3p and miR-429 Do Not Affect Neutrophil Migration

To assess whether miR-181b-3p/miR-429 affect neutrophil migration, static co-cultures were performed, where neutrophils were allowed to adhere and bind to HUVECs transfected with miR mimic, negative siRNA, lipofectamine only or optimem (Appendix A). There was no significant difference between groups, suggesting that the increased expression of miR-181b-3p/miR-429 did not have an impact on the transendothelial migration of neutrophils.

## 4. Discussion

This study is the first reported study on miR expression in pulmonary endothelial cells in patients with and without COPD and demonstrates that expression is significantly altered between groups. This may have relevance to cancer as miR changes in COPD patients may precede the development of lung cancer. One study demonstrated that the miR signatures (in blood) of COPD and lung cancer patients were more similar to each other than to controls, suggesting that some miR changes in lung cancer are already present in COPD [49].

This work is consistent with previous whole lung studies, which show clear differences in miR expression between patients with and without COPD [20,50]. There is a lack of evidence describing miR expression in pulmonary endothelial cells in COPD. However, previous studies have identified that some endothelial miRs are upregulated in COPD or on exposure to cigarette smoke. Several of these miRs were identified as upregulated in the microarray analysis in this study, supporting the study results (see Appendix A).

For example, Serban et al. described the enrichment of several miRs in EMPs released by cigarette smoke [22]. These miRs had putative functions of importance in COPD and cancer, such as endothelial activation and angiogenesis, airway inflammation and tumourigenesis [22]. One of these miRs (miR-125a) was also identified as upregulated in this study [22]. Another study investigating miR expression in COPD whole lung identified two miR as upregulated in COPD (miR-199a-5p and miR-34a), which were also upregulated in this study [23]. Although these miR were identified in the whole lung, it is likely that they are relevant in pulmonary endothelial cells as transfecting these miR into HPECs led to a reduction in HIF-1α [23]. Similarly, Sun et al. showed that miR-206, another miR increased in COPD in this study, was upregulated in whole lung tissues from COPD patients and resulted in VEGF suppression, inducing HPEC apoptosis in vitro [51]. Likewise, miR-34a, which was upregulated in the microarray analysis, has previously been shown to induce apoptosis in HPECs exposed to smoke through targeting Notch-1 [24]. It is also possible that some of the miRs identified as upregulated in this study could contribute to the vascular remodelling seen in COPD. Musri et al. identified several miRs with altered expression in COPD pulmonary arteries, including miR-146 and miR-451, which were upregulated in COPD in this microarray analysis [52]. 

In contrast to this study, other studies have demonstrated the suppression of some miRs in COPD. For example, Paschalaki et al. showed that miR-126 is suppressed in endothelial cells in COPD, although this was in blood outgrowth endothelial cells (BOECs) as opposed to HPECs [53]. Similarly, Musri et al. also discovered four miRs that were downregulated in COPD, although this was in pulmonary artery specimens [52].

This study has identified three targets which are upregulated in COPD endothelial cells: miR-181b-3p, -429 and -23c. These targets were not identified in previous COPD microarray studies [20,50], but this is not unexpected as results from this study are endothelial cell-specific, whereas other studies looked at whole lung or other cell types. 

There is little evidence about the function of miR-181b-3p. However, this miR has been shown to upregulate epithelial–mesenchymal transition (EMT) in breast cancer cells in vitro [54]. MiR-181b-3p levels are also associated with lymph node disease and a poorer prognosis in oral squamous cell cancer (OSCC), another smoking-related cancer [55]. This is suggestive of a possible link between miR-181b-3p and lung cancer. Further support for this hypothesis comes from the evidence that miR-181b-5p (a related miR from the same precursor as miR-181b-3p) is upregulated in lung squamous cell carcinoma and adenocarcinoma [56,57]. One explanation for this relationship with cancer could be that miR-181b-3p increases inflammation. MiR-181b, the precursor to miR-181b-3p, increases nuclear factor kappa-light-chain-enhancer of activated B cells (NF-κB) activity [58]. NF-κB is a ubiquitous transcription factor that plays an important role in the regulation of immune responses and inflammation [58] and so may also be of importance in pulmonary inflammation and COPD. miR-181b is also increased in gingival tissue in periodontitis, an inflammatory dental condition associated with COPD [59,60]. However, the evidence for miR-181b’s involvement in malignancy and inflammation is inconsistent. For example, Yang et al. found reduced miR-181b expression in NSCLC tissue [61]. Additionally, despite evidence linking miR-181b-3p to cancer, this miR was not upregulated in lung cancer endothelial cells in this study. Consequently, this study suggests that upregulation of miR-181b-3p in endothelial tissue is an unlikely mechanism of increased cancer risk in COPD.

MiR-429 appears to be consistently increased in lung cancer [62]. It is possible that miR-429 may cause changes in cell behaviour by targeting and suppressing known tumour suppressors: phosphatase and tensin homolog (PTEN), a phosphatase involved in apoptosis, Ras association domain family member 8 (RASSF8), a protein that maintains adherent junction function, and TIMP metalloproteinase inhibitor (TIMP2), which suppresses metastasis [62]. In addition, miR-429 has also been shown to be upregulated in serum from patients with lung cancer compared to patients with COPD [63]. Perhaps, therefore, it is possible that miR-429 is upregulated in COPD initially and further increases in the expression of this miR contribute to the development of lung cancer in these patients. This is supported by results from the qPCR experiments, which show a 2-fold increase in miR-429 in COPD, but a 9-fold increase in lung cancer pulmonary endothelial cells (in comparison to non-COPD). MiR-429 may also play an important role in pulmonary inflammation. MiR-429 upregulation results in the production of pro-inflammatory cytokines through the targeting of *DUSP1* (dual-specificity phosphatase 1) [64]. This may therefore have an important impact in COPD, in which the levels of pro-inflammatory cytokines are high.

The available information on miR-23c is limited. Thus far, the only study in humans that looked at the expression of miR-23c showed an upregulation of miR-23c in prostate cancer chemoresistant cells in comparison to cells that were chemosensitive [65]. MiR-23c also appears to target the tumour suppressor PTEN in a similar way to miR-429 and may therefore play a role in tumourigenesis [65]. However, miR-23c did not seem to be expressed in lung cancer endothelial cells and, therefore, it is less likely to provide an endothelial mechanism for the common pathogenesis of COPD and lung cancer.

Functional work demonstrated that miR-181b-3p and -429 overexpression significantly impairs tube formation and sprouting of endothelial cells in vitro, suggesting that these miRs impair angiogenesis. Wound closure, however, does not appear to be influenced by miR-181b-3p or -429. This is unexpected due to the consistent reduction in tube formation and endothelial sprouting seen when these miRs are overexpressed. However, scratch wound assays primarily assess cell migration and proliferation, but do not assess the same number of functions as the Matrigel, spheroid and endothelial–fibroblast co-culture assays [66]. Therefore, this could suggest that the reduction in angiogenesis is not primarily as a result of changes in cell migration; instead, it could be due to cell alignment and endothelial sprout formation. The lack of change seen in cell proliferation in the scratch wound assay supports the cell growth experiments, which were non-significant between miR overexpression and control siRNA groups.

Previous studies have shown a reduction in angiogenesis secondary to miRs related to miR-181b-3p. For example, in mouse models, injection of glioma cells expressing miR-181b mixed with Matrigel resulted in reduced angiogenic responses in the Matrigel plugs [67]. Results from another study appear to contradict this, however. One group identified that miR-181b was increased in retinoblastoma cells and enhanced angiogenesis through targeting programmed cell death-10 (PDCD10) and GATA binding protein 6 (GATA6) [68]. There are no other studies looking at angiogenesis in relation to miR-181b-3p as yet and thus it is difficult to draw firm conclusions as a result of the evidence available.

There is also evidence that miR-429 may influence angiogenesis. MiR-429 appears to induce apoptosis in endothelial cells by targeting Bcl-2, an anti-apoptotic protein [44]. In vivo studies using mouse models have demonstrated that miR-429 is upregulated in the aortic endothelial cells of mice with atherosclerosis and that this is associated with endothelial apoptosis [44]. Further evidence also supports the suppression of angiogenesis by miR-429 through HIF-1α and, consequently, VEGF suppression in vitro [69]. If miR-429 has a similar function in pulmonary endothelial cells, it is possible that this could result in pulmonary endothelial apoptosis and emphysema.

It has been postulated that the pulmonary microvascular circulation is important in the maintenance of alveolar structures by producing factors termed ‘angiocrines’, such as retinoic acid. There is evidence that these angiocrines may be reduced in emphysema. For example, patients with emphysema have increased levels of CYP26A1 (an enzyme that degrades retinoic acid) in the endothelium [70]. A reduction in angiocrines could potentially result in dysregulated maintenance and repair of alveoli, resulting in alveolar damage and emphysema. Preserving the lung vasculature to promote growth and preserve the alveolar architecture could be a treatment strategy in COPD. In support of this, murine models have demonstrated that retinoic acid can enhance lung growth after pneumonectomy [71]. These findings suggest that the lung may have more intrinsic regenerative ability than previously thought. As it has been shown that miRs increased in COPD can result in HPEC apoptosis through VEGF suppression [51], perhaps by correcting miR expression in COPD, we could limit the vascular degeneration and alveolar damage.

The endothelium also plays an important role in transendothelial migration (TEM), the process by which inflammatory cells such as the neutrophil cross the endothelial barrier into the lung [72]. As inflammation is associated with cancer, it is possible that dysregulation of the endothelium, resulting in upregulation of TEM and inflammation, could promote the development of cancer in the COPD lung. The fact that two of the identified endothelial targets in this study alter endothelial function implies that the miR changes seen in COPD pulmonary endothelium may have functional effects. As miR-429 was also upregulated in lung cancer, this suggests that changes in the COPD pulmonary endothelium might represent a pre-malignant state that can be targeted. However, the static adhesion assays performed in this study do not support miR-181b-3p or -429 acting via TEM.

The main strength of this study was the cell-specific analyses. This is the first study to report miR expression changes in human pulmonary endothelial cells. Very few studies have looked at the miR expression in COPD in individual cell types in the lung; this is important for two main reasons. Firstly, one cannot assume that changes in the whole lung apply to each cell type, which makes it difficult to identify potential cellular pathogenic pathways in COPD. Secondly, signals from different cells might cancel each other out, resulting in important targets being missed.

A major limitation was the size of the cohort and need for replication within it, which lowered the statistical power and means that additional endothelial signals cannot be excluded. This does not diminish the significance of the signals found, given that they were supported by functional data. The composition of the cohort might also have affected the microarray analyses; most patients had lung cancer so expression in the non-COPD group could be different compared to patients without lung cancer. In addition, patients in the COPD group primarily had mild–moderate disease and so expression could differ from severe COPD. However, these limitations are difficult to overcome given that lung resection outside of a clinical need would be unethical and that resection in severe disease, outside the context of transplantation or lung volume reduction, is not common.

## 5. Conclusions

MiR-driven changes in the pulmonary endothelium represent a novel mechanism driving pulmonary emphysema through alterations in angiogenesis. Targeting and normalising the levels of the aforementioned endothelial targets could provide a new mechanism of treatment for COPD that is focused on the repair and regeneration of lung tissue.

## Figures and Tables

**Figure 1 biomedicines-09-00830-f001:**
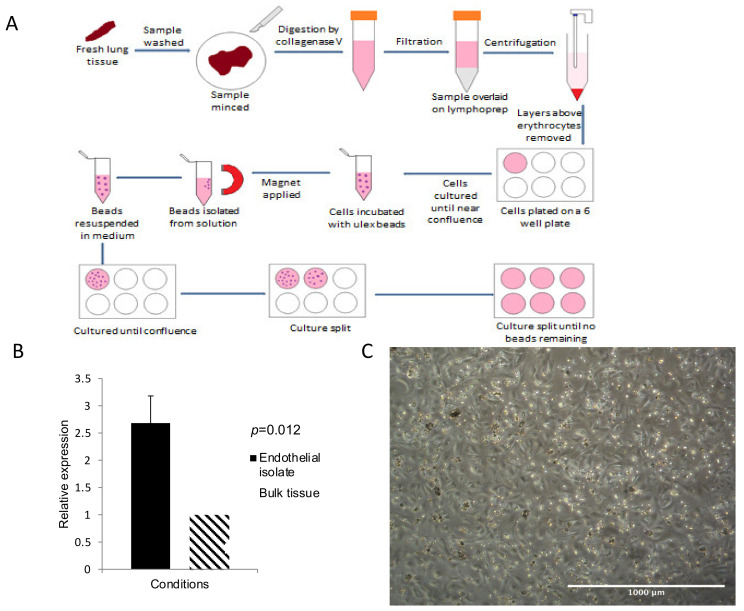
Isolation and culture of endothelial cells. (**A**) Workflow: Fresh lung is minced before being added to collagenase V, the digested sample is passed through a filter before being overlaid on lymphoprep, cell layers above the erythrocytes are removed and the cell pellet resuspended and placed in a 6-well plate. The cell culture is then incubated with Ulex-coated magnetic beads, which bind to endothelial cells, before a magnet is used to isolate the beads. Cells are used either for RNA extraction at this point, or the beads are resuspended and cultured by placement in a 6-well plate, the culture being split repeatedly until no beads remain. (**B**) Confirmation of isolation efficiency by qPCR: the bar chart represents the relative expression of CD31 in endothelial isolates in comparison to bulk tissue, normalised against flotillin 2. (**C**) Appearance of mixed lung cell culture on day five, extraction four. Primarily endothelial cells are seen, with some clumps of other cells. Scale bar represents 1000 μm.

**Figure 2 biomedicines-09-00830-f002:**
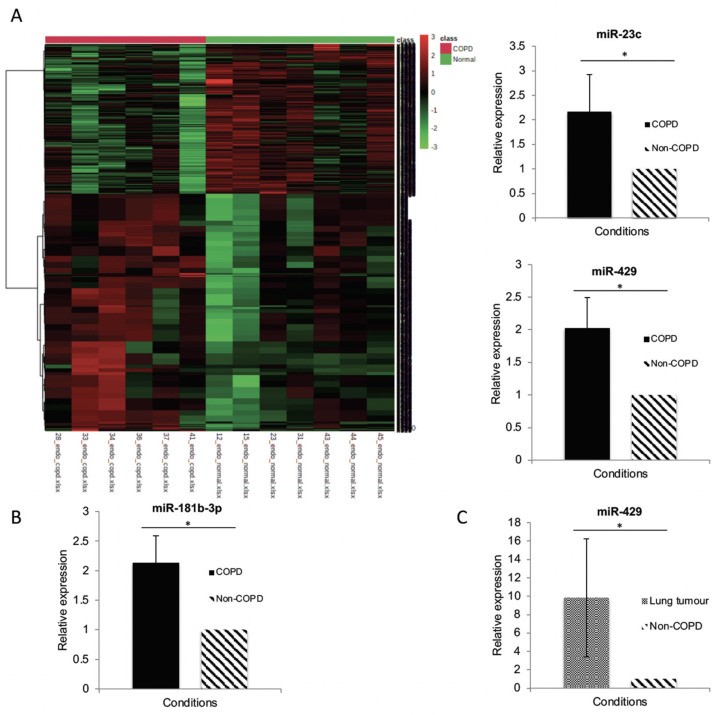
miR expression analyses. (**A**) Heatmap showing differentially expressed miR between COPD and non-COPD: this map comes from the final miR microarray analysis and is limited to the top 1000 miR significantly different in the test array. (**B**) qPCR validation of miR in COPD vs. non-COPD: bar charts represent relative expression for a miR. RNU48 was used as the house-keeping small RNA to which the data were normalised. The double delta CT method was used to compare expression levels; the 3 miR shown were significant at *p* <0.05. (*n* = 4 both groups). (**C**) qPCR validation of miR in cancer vs. non-cancer tissue: miR-429 was the only miR that was also significantly altered in tumour (*n* = 6) vs. normal lung (*n* = 4) (*p* = 0.017); relative expression was also higher than COPD lung. * = *p* < 0.05.

**Figure 3 biomedicines-09-00830-f003:**
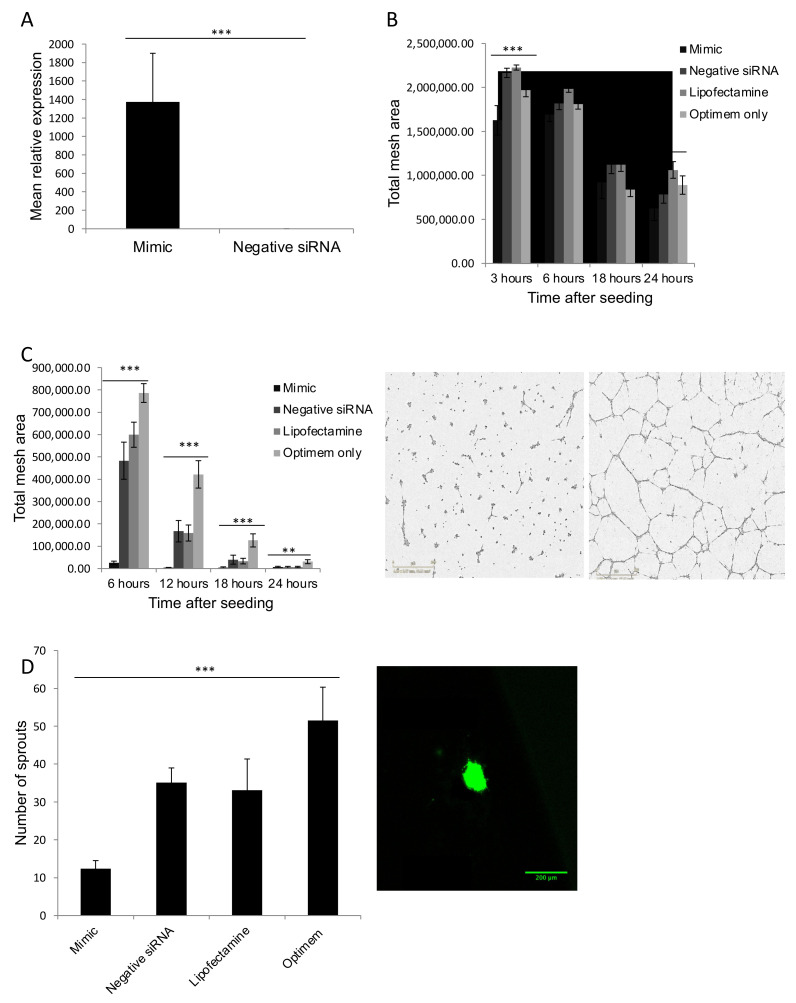
MiR overexpression in endothelial tissue. (**A**) Transfection of HUVEC cells with miR mimic: Since miR-181/-429 were upregulated in COPD lung, a mimic was used to test function. The mimic successfully upregulated the miR in HUVECs 3 days after transfection (*n* = 3 in both groups), *** = *p* < 0.001. (**B**) Number of nodes formed by HUVECs imbedded in Matrigel after transfection of miR-429 mimic. The figure shows the number of nodes present 3, 6, 18 and 24 h after HUVECs were seeded on Matrigel. The numbers represent the mean from 3 separate experiments. An ANOVA test was used to look for significance between groups. * = *p* < 0.05 (ANOVA); *** = *p* <0.001. Tukey’s test was used to compare groups, which demonstrated that number of nodes was significantly reduced in the mimic group in comparison to the negative siRNA group at 3 h (*p* < 0.001), suggesting that miR-429 reduces angiogenesis. (**C**) Total mesh area formed by HUVECs embedded in Matrigel after transfection of miR-181b-3p mimic: the figure shows the total mesh area present 6, 12, 18 and 24 h after HUVECs were seeded on Matrigel. The numbers represent the mean from 3 separate experiments. An ANOVA test was used to look for significance between groups. ** = *p* < 0.01 (ANOVA); *** = *p* <0.001. Tukey’s test was used to compare groups, which demonstrated that total mesh area was significantly reduced in the mimic group in comparison to the negative siRNA group at 6 (*p* < 0.01 and 12 h (*p* = 0.042), suggesting that miR-181b-3p reduces angiogenesis. Representative images from these experiments are shown; the picture on the left shows HUVECs at 12 h transfected with the miR mimic, and on the right with the negative siRNA. Scale bars represent 800 μm. (**D**) Number of endothelial sprouts from spheroids after transfection of a miR-181b-3p mimic: the figure shows the total number of endothelial sprouts for each condition. The numbers represent the mean from 3 separate experiments. An ANOVA test was used to look for significance between groups. *** = *p* <0.001 (ANOVA). Tukey’s test was used to compare groups, which demonstrated that number of sprouts was significantly reduced in the mimic group in comparison to the negative siRNA group. Representative images from these experiments are shown; the picture on the left shows HUVECs transfected with the miR mimic, and on the right with the negative siRNA. Scale bars represent 200 μm.

**Figure 4 biomedicines-09-00830-f004:**
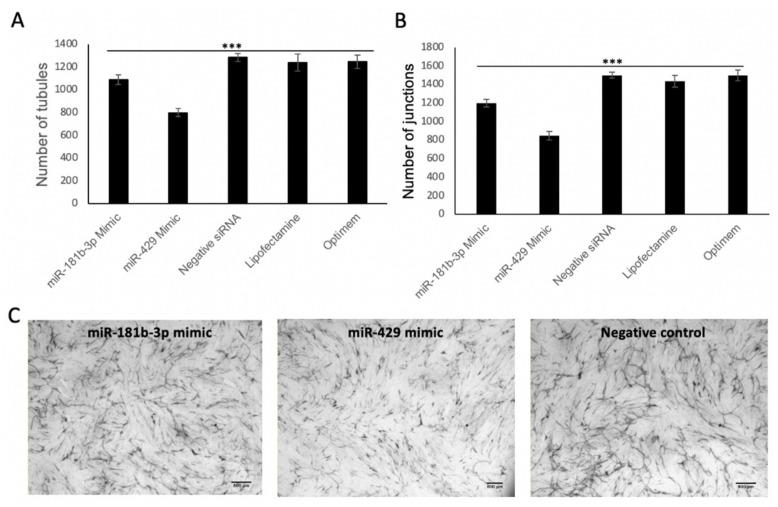
Endothelial–fibroblast co-cultures. (**A**) Number of tubules formed by HUVECs grown in co-culture with fibroblasts after transfection of miR-429 or miR-181b-3p mimics. The numbers represent the mean from 4 separate experiments. An ANOVA test was used to look for significance between groups. *** = *p* < 0.001. Tukey’s test was used to compare groups, which demonstrated that number of tubules was significantly reduced in both mimic groups in comparison to the negative siRNA group, suggesting that miR-429 and miR-181-3p reduce angiogenesis. (**B**) Number of junctions formed by HUVECs grown in co-culture with fibroblasts after transfection of miR-429 or miR-181b-3p mimics. The numbers represent the mean from 4 separate experiments. An ANOVA test was used to look for significance between groups. *** = *p* < 0.001. Tukey’s test was used to compare groups, which demonstrated that number of junctions was significantly reduced in both mimic groups in comparison to the negative siRNA group, suggesting that miR-429 and miR-181-3p reduce angiogenesis. (**C**) Representative images from these experiments are shown. Branching structures of HUVECs are seen, which are endothelial tubules. A reduction in tubule formation is seen in endothelial cells transfected with miR-181b-3p and miR-429 in comparison to transfection with the negative control. Scale bars represent 800 μm.

**Table 1 biomedicines-09-00830-t001:** Characteristics of the patient cohort from the miR microarray analysis.

Variable	All *n* = 13	Non-COPD *n* = 7	COPD *n* = 6	*p* Value
Male patients	5 (38.5%)	3 (42.9%)	2 (33.3%)	1.000
Age	67.0 (14.0)	63.0 (18.0)	69.0 (11.0)	0.366
BMI	21.8 (7.1)	27.2 (13.9)	21.6 (5.8)	0.731
Pack years	26.5 (40.5)	0 (31)	26.0 (61.3)	0.073
Current smoker	2 (15.4%)	1 (14.3%)	1 (16.7%)	0.086
FEV1pp	85.1 (36.0)	96.8 (20.4)	64.0 (40.4)	0.008
No cancer *	2 (15.4%)	0	2 (33.3%)	0.874
0 (in situ)	0	0	0	
IA	3 (23.1%)	2 (28.6%)	1 (16.7%)	
IB	3 (23.1%)	2 (28.6%)	1 (16.7%)	
IIA	0	0	0	
IIB	2 (15.4%)	1 (14.3%)	1 (16.7%)	
IIIA	2 (15.4%)	1 (14.3%)	1 (16.7%)	
Lung metastases	1 (7.7%)	1 (14.3%)	0	

Continuous variables are presented as median (interquartile range). Categorical variables are presented with percentages. Significant *p* values are presented in bold. * LVRS specimens or cases with benign tumours.

## Data Availability

All array data have been deposited in the ArrayExpress database at EMBL-EBI (www.ebi.ac.uk/arrayexpress) under the accession number E-MTAB-10311 [31].

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
