# Peer review of "Pulmonary MicroRNA Changes Alter Angiogenesis in Chronic Obstructive Pulmonary Disease and Lung Cancer"

_biomedicines, 2021, doi:10.3390/biomedicines9070830_

Round 1
Reviewer 1 Report
The Authors presented original data on the differential expression of a set of microRNAs in human pulmonary endothelial cells freshly isolated from lung tissues of lung cancer patients affected or not by COPD. They also demonstrate the functional role of miRNAs in a model of endothelial cells derived from umbelical vein (HUVEC). Representative images of HPEC should be shown in figure 1. The abstract should state that functional assays were performed using the HUVEC cell line model.
Author Response
The Authors presented original data on the differential expression of a set of microRNAs in human pulmonary endothelial cells freshly isolated from lung tissues of lung cancer patients affected or not by COPD. They also demonstrate the functional role of miRNAs in a model of endothelial cells derived from umbelical vein (HUVEC).
Representative images of HPEC should be shown in figure 1.
Thank you – we have done this.
The abstract should state that functional assays were performed using the HUVEC cell line model.
Thank you for your comment. We have altered the abstract to include this.

Reviewer 2 Report
An interesting and novel work. The small size of subjects is indeed a limitation as it has already been said by the authors. More tissue samples are at need for more conclusions.
Could you elaborate upon any other methods or results to be compared such as any immunocytochemistry procedures? Why did you prefer this method?
Also, discussion section should be in full and not seperated.
Author Response
Could you elaborate upon any other methods or results to be compared such as any immunocytochemistry procedures? Why did you prefer this method?
Thank you for your comment. We decided to focus on cell culture models rather than immunohistochemistry as the optimal way of investigating microRNA function as opposed to comparing microRNA expression between groups which had been investigated in the microarray and qPCR experiments. We focused on cell culture methods that had previously been used and published in previous works of endothelial function as these methods have previously been validated. Previous examples of work using similar methods include the following (these are also referenced in the text):
Zhuang, X., et al., Br J Cancer, 2015. 112(3): p. 485-94.
Noy, P.J., et al., Oncogene, 2015. 34(47): p. 5821-31.
Munir, H., et al., J Vis Exp, 2015(95): p. e52480.
Also, discussion section should be in full and not separated.
Thank you – we have made this alteration.
